# Music Emotion Recognition Based on a Neural Network with an Inception-GRU Residual Structure

**Xiao Han [1], Fuyang Chen [1,\*] and Junrong Ban [2]**

1   College of Automation Engineering, Nanjing University of Aeronautics and Astronautics,
    Nanjing 210016, China
2   College of Art, Nanjing University of Aeronautics and Astronautics, Nanjing 210016, China
\*   Correspondence: chenfuyang@nuaa.edu.cn; Tel.: +86-025-84892305-801

**Abstract:** As a key field in music information retrieval, music emotion recognition is indeed a challenging task. To enhance the accuracy of music emotion classification and recognition, this paper uses the idea of inception structure to use different receptive fields to extract features of different dimensions and perform compression, expansion, and recompression operations to mine more effective features and connect the timing signals in the residual network to the GRU module to extract timing features. A one-dimensional (1D) residual Convolutional Neural Network (CNN) with an improved Inception module and Gate Recurrent Unit (GRU) was presented and tested on the Soundtrack dataset. Fast Fourier Transform (FFT) was used to process the samples experimentally and determine their spectral characteristics. Compared with the shallow learning methods such as support vector machine and random forest and the deep learning method based on Visual Geometry Group (VGG) CNN proposed by Sarkar et al., the proposed deep learning method of the 1D CNN with the Inception-GRU residual structure demonstrated better performance in music emotion recognition and classification tasks, achieving an accuracy of 84%.

**Keywords:** music emotion recognition; neural network; machine learning; inception-GRU residual structure

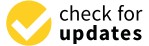



## 1. Introduction

As a crucial branch and key means of Music Information Retrieval (MIR), deep learning-based Music Emotion Recognition and Classification (MER) has become an active research field for MIR tasks [1,2]. When faced with a vast digital media library, it is difficult for people to find music works to review their listening preferences. Automatic music retrieval systems for different races, ethnicities, religions, age groups, and purposes are increasingly important. Among them, music emotion retrieval is an essential method of music retrieval, closely related to nationality, age group, mood, purpose, and cultural background. Different nationalities have different song styles, and different age groups have different needs for music. People from different cultural backgrounds have different needs for music styles (classical, pop, rock, etc.). Music has different uses in education, psychological counseling, music therapy, etc., and the purpose of emotional retrieval is also different according to different uses [3]. The perception of music emotion is not only subjective but also closely related to individual characteristics such as age, occupation, family environment, and background. These factors enhance the complexity of MER. With the emergence of digital music and the expansion of digital music libraries, manual retrieval and classification has become impossible, so there is an urgent need for automatic retrieval tools to automatically detect and classify attributes such as music genres, singers, and music emotions. Music is closely related to human emotions; listeners prefer to choose their favorite songs depending on their emotions. According to the need to express different musical emotions, composers will create a series of musical works expressing other feelings. These pieces of music bring not only appropriate musical expression to different emotional

backgrounds but also bring auditory enjoyment to the audience and even help to express the emotions of movies and TV works and describe stories. In terms of psychotherapy, appropriate music can soothe the inner trauma of patients. The choice of this music often does not care about the style and age. However, it pays more attention to the emotional expression in the music [4]. Different musical emotions release neurotransmitters and hormones (dopamine, serotonin, and oxytocin) and activate the reward and prosocial systems. Music affects functional brain connectivity and is effective in the treatment of neurological disorders. Mengru Sun proposed that music with different emotions can treat depression patients [5]. MER is not only applicable to music track navigation, search, and recommendation but is also widely used in the field of music therapy [6]. Therefore, MER is more useful than what appears to be in real-life scenes and has received wide recognition.

Conventional machine learning methods directly feed the labeled audio signal features into machine learning classifiers [7–9] and derive the classification results from low-dimensional features. This approach is not sufficient for sample training and limits classification accuracy. Accompanying effective applications in the field of computer vision and natural language processing, deep learning has also been adopted in the field of MER.

In this study, deep learning is combined with the four-quadrant emotion model proposed by Thayer [10] and Russell [11] to categorize music emotions into four categories: happy, angry, sad, and neutral. Based on this, a one-dimensional (1D) Convolutional Neural Network (CNN) with a residual structure containing the improved Inception module and Gate Recurrent Unit (GRU) for deep feature extraction is proposed. Besides, the efficiency of feature extraction is improved, and the accuracy of MER is enhanced. The system is evaluated on the popular music dataset Soundtrack.

The contribution of this work lies in the use of the optimized InceptionV1 module to enhance the feature extraction efficiency and classification accuracy by combining the 1D neural network with the GRU network in the residual structure. After pre-processing, the audio clips are fed into the deep learning system for emotion classification. Section 2 explores the related work on short-time Fourier transform (STFT); Section 3 describes the proposed method in detail; and Sections 4 and 5 highlight the experimental results and summary, respectively.

A Batch Normalization (BN) layer is added to this module to suppress over-fitting and accelerate the convergence of the training [12]. Fused with the convolutional layer, a feature expansion-compression-expansion structure is developed, which preserves the key features while guaranteeing the mining of valid information. Meanwhile, multiple expansion–compression–expansion structures are used parallelly to extend the diversity of extracted features. The problems of gradient vanishing and network degradation in deep convolutional networks are addressed by the residual structures. By adding GRU, which is easier to compute than the Long Short-Term Memory (LSTM) before the Softmax layer, the sequence data can be processed, and the problems of gradient exploding and vanishing can be solved in the long sequence data processing.

## 2. Related Work

Music is the language of emotions. In recent years, music emotion recognition has attracted widespread attention. With the rapid development of artificial intelligence, deep learning-based music emotion recognition is gradually becoming mainstream [13]. Music is capable of conveying many emotions. The level and type of emotion of the music perceived by a listener, however, is highly subjective. MER belongs to the interdisciplinary research field of music psychology, audio signal processing, and natural language processing (NLP). MER is a sub-task of music information retrieval (MIR). MER can be widely used in many fields, including music recommendation, retrieval, visualization, automatic music composing, psychotherapy, etc. Therefore, MER has become a research hotspot in the academic and industrial community [14]. Music emotion recognition has become an active research field in MIR tasks. Only two primary steps are required to achieve emotion recognition and classification: designing appropriate emotion features for the audio signals

and feeding the signals in the machine learning model for feature extraction and clustering. In the four-quadrant emotion model by Thayer [10] and Russell [11], the music features were employed using regression approaches for classification, whereas the other methods used classifiers in machine learning for recognition. In this section, the feature processing and recognition methods in music emotion recognition are briefly explained.

Many inherent attributes in music can be associated with music emotions, so a variety of manual feature labels have been designed for emotion classification. Among them, the speed and rhythm characteristics of music are often used to distinguish emotions. Fast music clips are often associated with positive valence, while slow music clips are often associated with negative valence [15]. Researchers also used techniques such as Mel Frequency Cepstrum Coefficient (MFCC) [16], Zero Crossing Rate (ZCR) [17], and pitch analysis [18] to design the timbre features.

The manual design of feature labels and feature extractors is complex and time-consuming; therefore, the use of deep learning for effective feature extraction becomes more urgent. Deep learning methods have been adopted in the fields of face recognition, natural language processing, and video processing. Deep learning methods are also used to process acoustic audio signals for linguistic emotion analysis [2]. In the field of MIR, although used less often, deep learning is a major trend in the face of massive music databases. Many researchers have used CNN and Recurrent Neural Networks (RNN) widely, which chiefly iterated the audio signal characteristics. By layer-by-layer information accumulation and optimization, the best output data that could characterize the input audio signal was obtained, which enabled the deep learning model to learn the desired music emotion information from the audio samples. Thus, accurate mathematical models were built to enhance the accuracy of music emotion recognition.

Regression-based methods categorized emotions into four quadrants in two-dimensional space [19]. The *x*-axis and *y*-axis represented arousal and valence, respectively, forming a four-quadrant emotion model. Russell considered arousal and valence to be the core of music emotions; arousal represented the strength and energy of music, while valence reflects the pleasure of music to people, as represented in Figure 1.

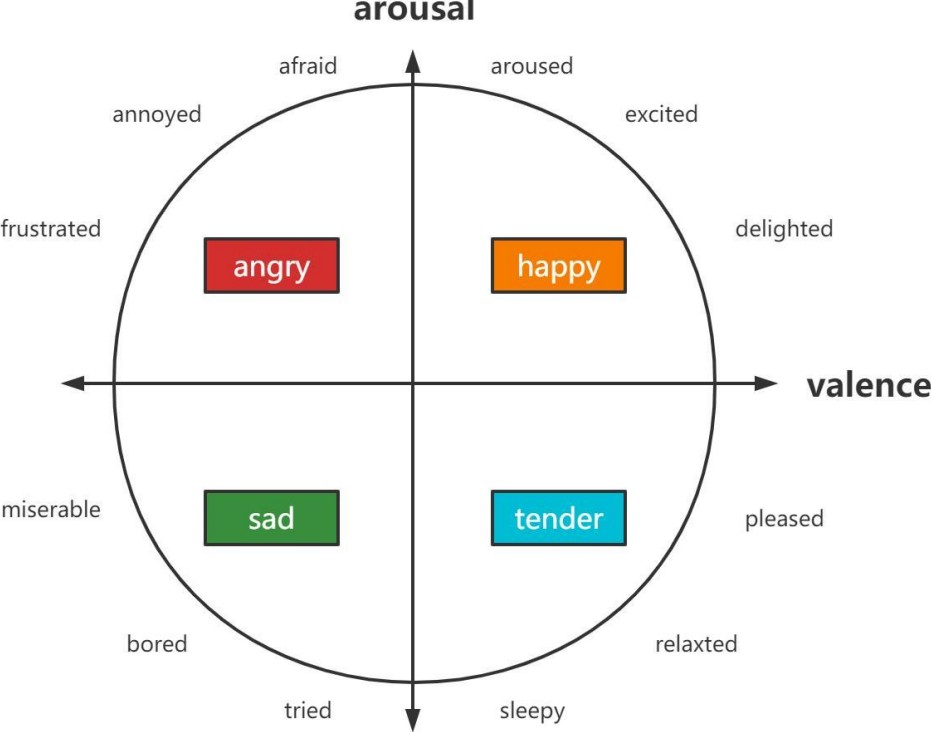

**Figure 1.** Two-dimensional emotion plane: valence vs. arousal.

In the proposed regression method, the label of music clips in the training set was a two-dimensional value of valence and arousal applicable to the four-quadrant emotion model. Observation values were formed from low-level features, and the regression model was formed to predict valence as well as arousal. A single regression model might be trained in terms of valence and arousal [18]. In the two-dimensional coordinate system formed by the valence and arousal values, the four quadrants formed the four basic emotions, and the valence and arousal scores of the auditory perception score jointly determine the various emotions of the music. Different emotions will be attributed to the four basic emotions according to the different quadrants. Researchers applied various methods to predict valence and arousal values, e.g., using Multiple Linear Regression (MLR), Support Vector Regression (SVR), and AdaBoost [20] on the feature set of spectral contrast images for audio. Han et al. [21] used SVR to extract the characteristics of pitch, rhythm, and harmonic to determine the emotion of music. Among the methods based on deep learning classifiers, the commonly used classifiers included Support Vector Machine (SVM) [9], Artificial Neural Network (ANN), Gaussian Mixed Model (GMM), and random forest. Eunjeong Koh et al. used VGGish and $L^3$-Net methods in the feature extraction process to classify music emotions using different classifiers [22]. Over-fitting, gradient vanishing, and gradient exploding might occur while using these methods, so researchers used dropout, regularization, or gradient shearing to adjust the machine learning framework.

Researchers have applied various regression frameworks and classifier models to the field of music emotion recognition; however, there is still room for improvement in classification accuracy and network structure rationality, so music emotion recognition is still a field worth studying.

## 3. Methodology

Since musical emotions are subjective and highly related to psychological problems, the recognition of musical emotions is a complex task [14]. Features labeled by non-automated design are shallow and intuitive, leading to limited sample processing power, incomplete feature extraction, and slow feature extraction speed, significantly affecting the classifier performance. These defects prompt the use of deep learning strategies to improve the feature extraction efficiency and classifier classification accuracy. The processed audio signal is fed into the deep learning network, and the best output data that can characterize the emotional factors of the input audio signal is determined through layer-by-layer information accumulation and optimization.

The method used in this paper is primarily divided into three steps. As shown in Figure 2, First, the audio signals in the dataset are pre-processed and converted into input data forms suitable for the deep learning system. The neural network then predicts the output of the input data and eventually processes the output data for emotion prediction.

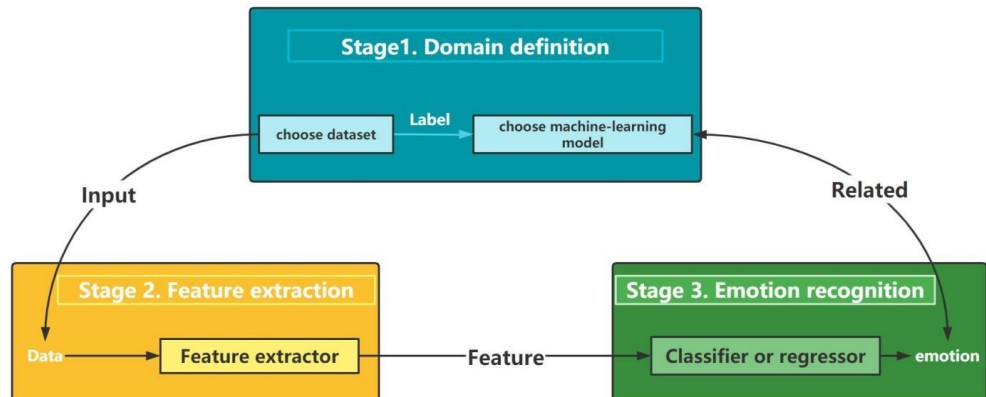

**Figure 2.** Three steps of the proposed method.

### 3.1. Pre-Processing

The original audio signal must be pre-processed before entering the deep learning network. First, the original music is cut into several music clips of 30 s, each being the basic unit of music emotion recognition. If the music clip is shorter than 30 s, it should be complemented to 30 s by copying the original audio using an audio editing tool.

STFT is used to process the music clips. Before Fourier transform, the audio signals are multiplied with a window function for framing and windowing. Meanwhile, the non-stationary audio signal is considered as an infinite number of stationary signals within a short period, and the FFT of each frame is performed. By moving the window function on the timeline, the spectrograms of the frames can be stacked on the timeline. The result is a Two-dimensional spectrogram with three-dimensional information highlighting various frequency states at different time points. Since CNN has exhibited commendable feature extraction performance in image processing, the time-frequency maps can be used as the data input of CNN to convert audio information to image information for processing, thereby providing full play to the merits of CNN in image processing. After using the STFT method for speech signal processing, the relationship diagram between energy and signal frequency can be generated, which are phase spectrogram and amplitude spectrogram, respectively. The STFT method can create a power spectrogram showing the change of signal power with frequency; it can also generate the spectrogram representing the three-dimensional information, which is the time, frequency, speech energy, and feature data used in this paper. The horizontal and vertical axes represent the sampling time (number of frames) and sampling frequency in the spectrogram. Each element in the spectrogram represents the energy of each frequency component in the music clip, which can characterize the depth of the color. The dimension of the spectrogram is $1025 \times 1292$, which serves as the input data for the neural network. The steps of pre-processing can be summarized as follows:

- Cut the audio into 30-s clips.
- Perform STFT for the music clips to produce two-dimensional spectrograms.
- Use the logarithm of the resulting spectrogram to highlight features and feed more prominent factors into the neural network.

### 3.2. Network Structure

An optimized Inception-V1 structure is proposed for data pre-processing, as illustrated in Figure 3.

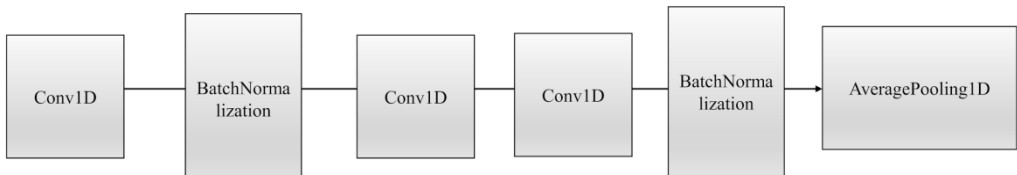

**Figure 3.** Optimized Inception-V1 structure.

Inception combines different convolution layers in parallel, and a deeper matrix is formed by concatenating the results processed by the convolution layers. The Inception module can be stacked repeatedly to form a larger network, which can effectively expand the depth and height of the network. While enhancing the accuracy of the deep learning network, visual information can be aggregated at different scales to facilitate feature extraction at varying scales. The traditional Inception-V1 structure is demonstrated in Figure 4.

The Inception module includes convolution operations. Unlike conventional CNNs, this module can set multiple channels that can involve different operations. Various convolution sizes provide different receptive fields, which allows feature extraction at different levels. The pooling operation itself has a feature extraction function and does not contain any parameters, so over-fitting can be eliminated. Therefore, the pooling operation also constitutes a channel of this module.

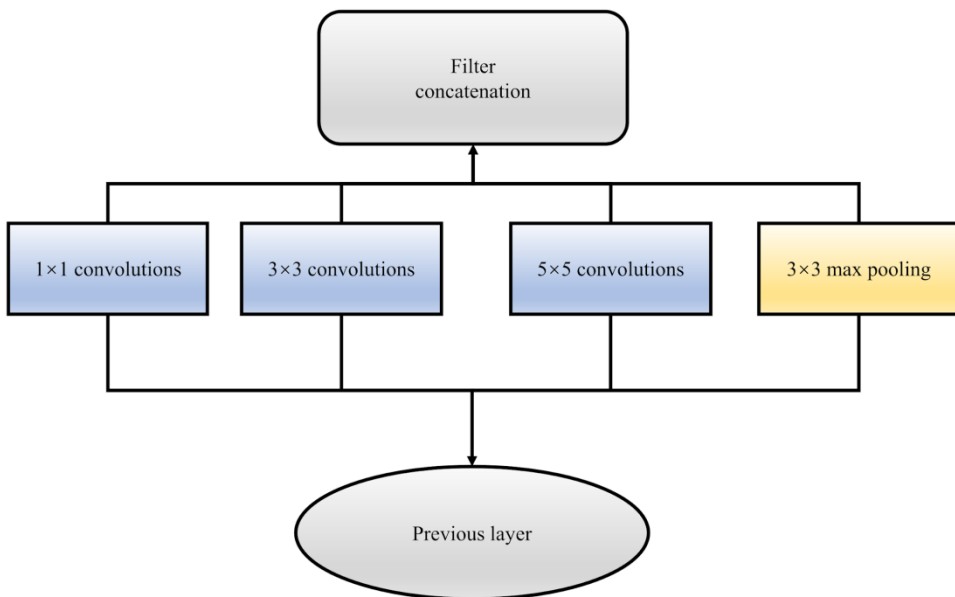

**Figure 4.** Traditional Inception-V1 structure.

　　CNN is a crucial neural network structure in deep learning, which is inspired by the receptive field mechanism in biology and can be adopted for processing similar network structure data such as time series data and image data. Its largest structural features include local connectivity, weight sharing, and temporal and spatial downsampling.

　　1D CNN is used by the optimized Inception module to perform 1D convolution along the width (time) direction using a two-dimensional convolution kernel on the input graph. Each convolution operation has a fixed number of convolution kernels that undergo characteristic mapping with the input data of the response layer. Here, $W_1$, $W_2$, $W_3$ ... denote convolution kernels with a length of K. The convolution sums of the convolution kernels and signal sequence $x_1$, $x_2$, $x_3$ ... is as follows:

$$y_t = \sum_{k=1}^{k} w_k x_{t-k+1} \tag{1}$$

where $y_t$ denotes the sum of the current and previous information of the audio signal received at moment $t$.

　　CNN is, in essence, an output-to-input mapping, which can learn several mapping relationships between input and output without requiring a precise expression between them.

　　If the convolutional network is trained using a known mode, it seeks the ability to map output to input. Since feature extraction of the convolutional network is learned from the training data, it avoids explicit feature extraction but implicitly learns from the training data to extract deep features of samples. Owing to the identical weights of the same feature mapping surface, parallel learning is possible. Depending on the special structure of weight sharing, the short- and long-term dependencies between features can be captured, making it extremely advantageous for audio processing.

　　Between the convolution layers of the optimized Inception module, the BN layer is added to normalize the data. This layer not only accelerates the convergence speed of the model but also alleviates the problem of gradient diffusion, making deep network training easier and more stable. BN consists of the following steps:

- obtain the mean of data in each training batch;
- obtain the variance of data in each training batch;
- normalize the batch of training data using the computed mean and variance to obtain a (0,1) normal distribution;
- conduct scale transformation and offset to obtain the distribution that best reflects the characteristics of the training samples.

Based on the one-dimensional convolutional neural network and the batch normalization layer, an optimized inception structure is formed to process the original features. This module refers to the Inception structure and uses convolution kernels of different sizes as receptive fields to extract features of different dimensions. The larger the value of the receptive field of a neuron, the more extensive the range of the original image it can touch, which also means that it may contain more features on a global level and a higher semantic level. The smaller the value, the more local and detailed the features it has, which increases the applicability of the network to different dimensions, makes the information extracted by feature complementary, and the feature extraction more efficient and comprehensive. In this structure, the convolutional neural network compresses and expands the features. During the compression process, the dimensionality of the features is reduced, the most representative information in the spectrogram is extracted, the amount of input information is reduced, and the reduced information is put into the neural network learning. This module facilitates the automatic learning burden of the neural network. Then upgrade the feature through feature expansion and then restore the feature to the original dimension to reconstruct the feature output, forming a compression-expansion-compression structure for the input feature. This process can effectively retain the main features while ensuring sufficient feature information. Mining, retaining as much information as possible, enables new representations with many different properties. The optimized Inception module is shown in Figure 5.

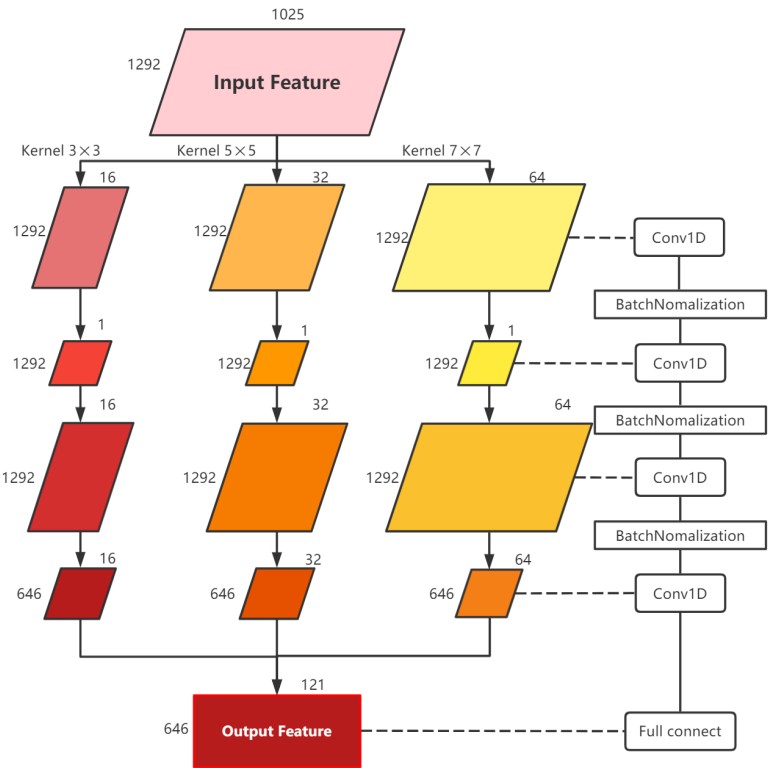

**Figure 5.** Optimized Inception module.

After data pre-processing is done by the optimized Inception module, the data is fed into the 1D convolutional network with the residual structures, as illustrated in Figure 6.

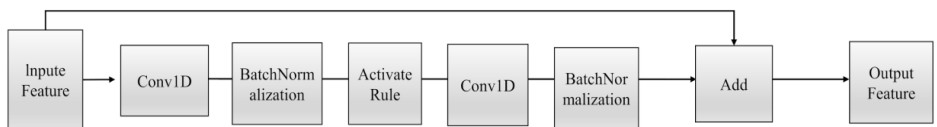

**Figure 6.** 1D Convolutional network with residual structure.

The residual module is composed of the key path and skip connection. The development trend of classical CNN is attained by stacking the layers. If too many layers are stacked, the problem of gradient vanishing might occur, which will eventually result in the failure of function convergence. Researchers have resorted to convergence by random gradient descent using normalized initialization and the intermediate normalization layer. As the number of network layers increases, the problem of network degradation emerges, leading to an increase in training error rate and a decrease in accuracy. To enhance the accuracy, the residual structure with the core being the residual block is introduced. In Residual Structure. $x$ is the input data, $H(x)$ is the output data, and the residual that the network must learn is $F(x) = H(x) - x$. Its original learning feature is $F(x) + x$.

The key idea is to add several identical mappings after a shallow network with a saturated accuracy. This approach increases the network depth without increasing the error, overcoming the constraints of the number of network layers and improving the accuracy.

After the 1D convolution network with the residual module, the data is fed into the GRU network. Since the audio data contains the characteristics of a time series, GRU can better capture the large-interval dependencies in the time series data. GRU is a type of RNN. In GRU network Structure, the transferred state $H_{t-1}$ and the output $x_t$ of the current node are used to achieve two gate states, i.e., the reset and update gates.

The reset gate is $r_t = \sigma(x_t W_{xr} + H_{t-1} W_{hr} + b_r)$; the update gate is $z_t = \sigma(x_t W_{xz} + H_{t-1} W_{hz} + b_z)$; and the candidate hidden layer state of the reset gate is $\widetilde{h} = tan\ tan\ h\ (x_t W_{hx} + r_t \odot h_{t-1} W_{hh} + b_h)$, where $h_{t-1}$ carries information about the past, $r_t$ denotes the reset gate and $\odot$ denotes multiplication by elements.

The final hidden state of the update gate is $h_t = (1 - z_t) \odot h_{t-1} + z_t \odot \widetilde{h}_t$, where $h_{t-1}$ contains information about the past, $\widetilde{h}_t$ represents the candidate's hidden state, and $z_t$ denotes the non-updated date. This step aims to forget some dimension information in $h_{t-1}$ that is passed down and add some dimension information entered by the current node. $(1 - z_t) \odot h_{t-1}$ indicates selective forgetting of the hidden state, and $z_t \odot \widetilde{h}_t$ denotes selective memory of the candidate hidden state of the current node. When $z_t$ approaches 1, long-term dependency persists. When $z_t$ reduces to 0, some unimportant information from the hidden information is forgotten.

Another contribution of this paper is to combine the residual structure with the GRU module, as shown in Figure 7, while solving the gradient disappearance and network degradation of the deep neural network. It can extract high-dimensional features and obtain higher classification accuracy. Moreover, the residual structure's features of different time series outputs are sent to the corresponding GRU unit. Finally, according to the characteristics of the audio data, the semantic association between long sequences is captured to capture the time series features in the audio data and further alleviate the problem of gradient disappearance in a better way.

Figure 8 illustrates the block diagram of the proposed neural network. The roles of the various layers are summarized as follows:

- 1D CNN: it is suitable for processing time series signals of music;
- adjusted Inception structure: It gets adjusted based on InceptionV1;
- the feature expansion-compression-expansion steps can effectively preserve the main features while guaranteeing the mining of valid feature information;
- parallel use of multiple optimized Inception structures to achieve multiple expansion-compression-expansion paths in parallel and extend the diversity of features;
- 1D residual structure: It avoids gradient vanishing of the deep network;
- combination with GRU model: It is used to deal with time series music signals and retain valid features through gating.

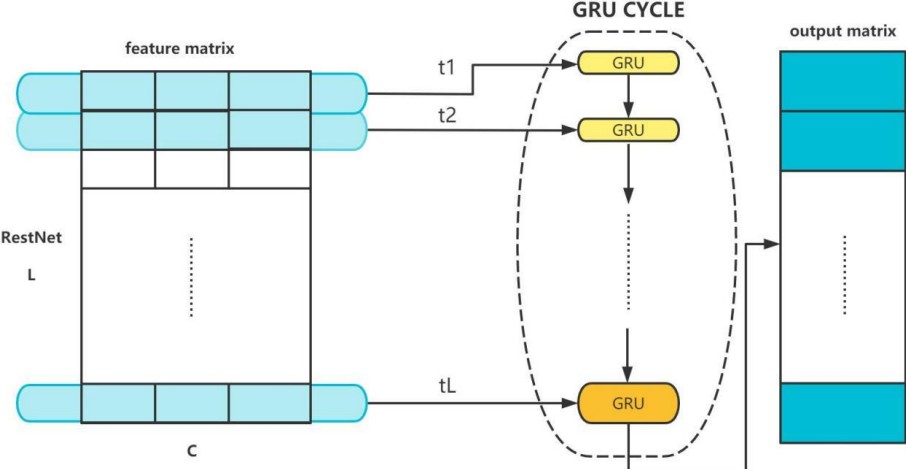

**Figure 7.** The timing signal through the residual network is connected to the GRU module.

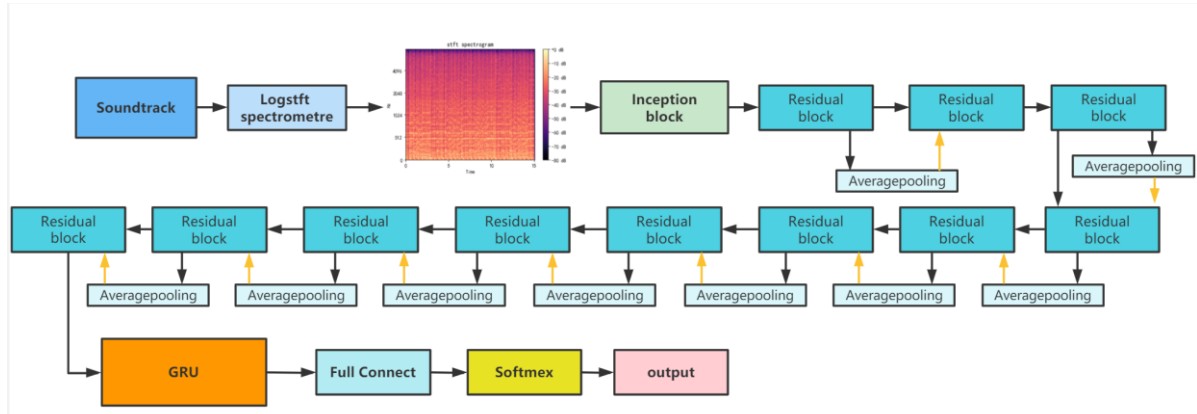

**Figure 8.** Block diagram of the proposed neural network.

A detailed framework of the neural network is represented in Table 1:

**Table 1.** The architecture of the proposed convolutional neural network.

| Layer | | Output Size | Filter_Number, Kernel Size |
|---|---|---|---|
| | Input | 1292, 1025 | - |
| | Conv1D | [1292, 16] [1292, 32] [1292, 64] | [16,3] [32,5] [64,7] |
| | BatchNormalization | [1292, 16] [1292, 32] [1292, 64] | - |
| | Conv1D | [1292, 1] [1292, 1] [1292, 1] | [1,3] [1,5] [1,7] |
| SE-Inception | Conv1D | [1292, 16] [1292, 32] [1292, 64] | [16,3] [32, 5] [64,7] |
| | BatchNormalization | [1292, 16] [1292, 32] [1292, 64] | - |
| | AveragePooling1D | [646, 16] [646, 32] [646, 64] | [2] [2] [2] |
| | Concatenate | 646, 112 | - |
| residual_1d_block × 2 | | 646, 112 | [112,3] |
| average_pooling1d | | 323, 112 | [2] |
| residual_1d_block × 2 | | 323, 112 | [112,3] |
| average_pooling1d | | 161, 112 | [2] |
| residual_1d_block × 2 | | 161, 112 | [161, 3] |
| average_pooling1d | | 80, 112 | [2] |
| residual_1d_block × 2 | | 80, 112 | [80, 3] |

**Table 1.** *Cont.*

| Layer | Output Size | Filter_Number, Kernel Size |
|---|---|---|
| average_pooling1d | 40, 112 | [2] |
| residual_1d_block | 40, 112 | [40, 3] |
| average_pooling1d | 20, 112 | [2] |
| residual_1d_block | 20, 112 | [20, 3] |
| average_pooling1d | 10, 112 | [2] |
| residual_1d_block | 10, 112 | [10, 3] |
| average_pooling1d | 5, 112 | [2] |
| residual_1d_block | 5, 112 | [5, 3] |
| average_pooling1d | 2, 112 | [2] |
| GUR | 8 | - |
| Dense_softmax | 4 | - |

## 4. Experiment Results

The experiment was performed on the Soundtrack [23] dataset based on audio feature processing and deep learning, as demonstrated in Figure 1. Four types of emotions were taken as the categories to be classified in the experiment, which corresponded to the four quadrant categories illustrated in Figure 1. The four emotions were happy, angry, sad, and tender, respectively. The Soundtrack [23] dataset contained 360 sound samples, each of which was a 30-s piece of music extracted from movie soundtracks. The STFT method used in the feature extraction generates a spectral feature map. The feature recognition of the spectral feature-driven image is used to associate the spectral features of the emotional information domain in the audio data to classify the music data set. The data set is based on the emotional arousal/valence two-dimensional emotional model data set. The arousal axis is rated from 1 to 8 from tender/sleepy to tension/exciting, and the valence axis is rated from 1 to 8 from sad/frustrated to happy/pleased. Every track has prominent emotional labels. Data emotion classification comes from the auditory perception of the human body and does not analyze its tonality from the note level. Still, it should be noted that the audio data comes from movie music with prominent emotional characteristics. The music's timbre is an instrumental performance without the features of human voice and lyrics text, and the audio was stored at 44.1 kHz sampling rate in mp3 stereo format. In the experiment, the maximum confidence label was used to match, with 156 audio clips labeled angry, 58 happy, 68 sad, and 78 tender. The ratio of the training set to the test set was 8:2. The experiment was concluded with the classification of cross-entropy between prediction and loss function. The optimization method used was Adam, and the learning rate was set to 1e-3. The code used by STFT was written in Python using the Librosa library [24], and the training model used the Keras library [25], also written in Python.

The precision, recall, and F-1 score of the proposed model on the Soundtrack dataset are mentioned in Table 2. Evidently, confusion appeared between the categories of sad and tender (the third and fourth quadrants in the four-quadrant emotion model presented by Thayer [10] and Russell [11]), as well as happy and angry (the first and second quadrants). This was possibly because both sad and tender categories belonged to the low arousal category of the two-dimensional Russell plane, whereas happy and angry belonged to the high arousal category. It can be inferred from the results that the model exhibited strong emotion discrimination based on arousal but relatively poor discrimination based on valence.

**Table 2.** Precision, Recall and F-1 score (in %).

| Class | Precision | Recall | F1-Score |
|---|---|---|---|
| Anger | 0.88 | 0.79 | 0.86 |
| Sad | 0.69 | 0.88 | 0.71 |
| Tender | 0.63 | 0.63 | 0.72 |
| Happy | 0.93 | 0.93 | 0.80 |

The proposed neural network model was compared with the recently proposed models, as represented in Table 3, including three deep learning models and three traditional machine learning classification methods. Saari et al. [26] achieved a 54% accuracy in music emotion classification on the Soundtrack dataset based on SVM and k-Nearest Neighbor (K-NN), which was not sufficient for MIR. With the development of deep learning networks, the Visual Geometry Group (VGG)-16 network [27] achieved a classification accuracy of 66%. The emergence of RNN enables the neural network to process time-domain features and has been widely used in natural language processing. The Multi-Channel Convolutional LSTM (MCCLSTM) network [28], a variant of RNN based on CNN and LSTM, exhibited 74.35% accuracy on this dataset. However, in the case of lower parameters, it was difficult to achieve a higher accuracy only by combining CNN with LSTM. Therefore, Sarkar et al. proposed an improved VGG network [29] with relatively few layers and achieved 82.54% accuracy on the Soundtrack dataset. Chaudhary et al. increased the accuracy to 83.98% by superposition of convolution kernels with different sizes [30], which was still lower than that of the proposed model. The experimental results reflected that the proposed network model achieved a higher accuracy.

**Table 3.** Comparison of performance.

| Method | Accuracy | Param | FLOPs | Recall | F1-Score |
|---|---|---|---|---|---|
| SVM + ReliefF | 54.63% | - | - | | |
| VGG-16 | 56.45% | - | - | 72.69% | 76.4% |
| Inception-v3 | 68.12% | 14,992,068 | 85,681,045,528 | | |
| ResNet-18 | 74.35% | 2,487,978 | 1,230,683,852 | | |
| Sarkar et al. [29]. | 82.54% | 1,203,140 | 17,710,549,784 | 61.87% | 62.66% |
| Chaudhary et al. [30]. | 83.98% | 375,012 | 5,922,046,296 | 74.53% | 69.46% |
| Proposed approach | 84.23% | | | 79.63% | 77.36% |

## 5. Conclusions

This paper provided a MIR network characterized by musical emotion, which used the spectral eigenvector matrix formed by logarithmic STFT in the time domain as the data input. The proposed neural network framework was found to be competitive on the Soundtrack dataset. The 1D CNN based on the optimized inception-GRU residual structure achieved an accuracy of 84.27%, which was significantly higher than the other recently proposed shallow and deep learning models for music emotion classification. The improved Inception module was optimized based on the structure of Inception-V1. The BN layer was added between the 1D convolution layers. The parallel expansion-compression-expansion steps of multiple feature input data effectively preserved the key features while guaranteeing the mining of valid feature information and expanding the diversity of the extracted features. The 1D convolution network with the residual structure combined with the GRU module avoided the gradient vanishing of the deep network and was more suitable for processing the time series characteristics of audio files. With the help of this model to build a music retrieval system, it can more accurately recommend different emotional music and other background music according to the needs of different emotional scenes. In music therapy, different songs can be recommended according to different conditions to adjust the physiological and psychological state to treat diseases. In the future, as the number of data increases or the sample time decreases, so the amount of data after STFT processing increases, the accuracy of the model can be further enhanced. It will also be worthwhile to analyze the transfer learning capability of this model.

**Author Contributions:** Conceptualization, X.H., F.C. and J.B.; methodology, X.H.; software, X.H.; validation, X.H., F.C. and J.B.; formal analysis, X.H.; investigation, X.H.; resources, F.C.; data curation, X.H.; writing—original draft preparation, X.H.; writing—review and editing, X.H.; visualization, X.H.; supervision, X.H.; All authors have read and agreed to the published version of the manuscript.

**Funding:** This research received no external funding.

**Data Availability Statement:** Not applicable.

**Conflicts of Interest:** The authors declare no conflict of interest.

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
