# Peer review of "Music Emotion Recognition Based on a Neural Network with an Inception-GRU Residual Structure"

_electronics, doi:10.3390/electronics12040978_

Round 1

Reviewer 1 Report

Authors in this paper provided a MIR network characterized by musical emotion, which used the spectral eigenvector matrix formed by logarithmic STFT in the time domain as the data input. In the paper the proposed neural network  was  competitive on the Soundtrack dataset. Authors claim that the 1D CNN based on the optimized inception-GRU residual structure achieved an accuracy of 84.27%, which was significantly higher than the other recently proposed shallow and deep learning models for music emotion classification that is described in the presented literature. 

Author Response

Please see the attachment, thank you!

Reviewer 2 Report

The authors should mention emotion model used in the present study. The following paper should be referred, doi:10.1007/s12021-022-09579-2. 

In particular, Figures 2,7,8,10 are useless. Please remove them. Indeed, a graphical abstract should be provided.

The parameters of the data should be clarified in terms of length (duration), tonality, power spectra, etc.. in accordance with emotional arousal/valance scores. 

Author Response

Please see the attachment, thanks a lot!

Reviewer 3 Report

Review

Music Emotion Recognition Based on Neural Network with an Inception-GRU Residual Structure

Thank you for the opportunity to review this manuscript. It is evident that authors are knowledge and experts in the area of using technology to explore music emotion recognition. I believe the manuscript has great value and will provide a significant contribution to the literature.

I have included feedback below to help strengthen the manuscript for publication. I encourage the authors to ensure that the literature is being utilized to support the claims and statements that are being made.

Page 1 Line 28: The authors mention that perception of music emotion is based on several factors including background. Background is rather vague and it would clarify for the reader if this was explicated more (cultural, education, etc.).

Page 1 Line 33: There are no sources cited related to listeners preference to listen to songs depending on their emotions. This is a key statement for the premise of the manuscript AND as a result it should be grounded in the literature.

Page 1 Line 34-35: The authors indicate that MER is widely applicable to the field of music therapy, however, no music therapy literature is cited as evidence. There is music therapy literature to support this, it is highly recommended the authors draw upon this literature to support this statement.

Page 2 Line 67: The authors state that music emotion recognition has become an active research field in MIR tasks. There are not sources to cite this statement. It is vital to ensure that statements like this are supported by the literature.

Page 3 Line 120: The authors restate that musical emotions are subjective and highly related to psychological problems. Again, these needs to be literature to support these statements.

Page 4 Line 129: It would be helpful to the reader if the authors would explicate the three steps more clearly.

In the introduction of the manuscript, the authors state that this research is widely applicable to music therapy, this is not addressed in the discussion or conclusion. While the authors have provided an extensive reporting of the data, returning to how this relates to and informs these disciplines is important.

Author Response

(The authors gave the same response as above.)

Reviewer 4 Report

The paper presents a system for music emotion recognition (MER) using a deep neural network architecture using an existing Inception structure. The authors seem to have done some optimization of the network, but it is not clear what this optimization is. Was the network retrained? where some of the parameters changed?  If the optimization means adapting the network to work on spectrograms, it need to be said explicitly and the wording should be made more precise. The details of the multi-channel network are missing. When the user mention STFT and then spectrogram, the phase information was apparently removed. Not all works in deep learning of audio ignore the phase, so the use of magnitude STFT should be addressed.

The novelty of the paper is mostly in the application of existing models to the MER problem. Most of the paper presents general and largely known details of existing deep learning networks in a tutorial style. This might not be necessary for a research paper and the paper can be edited with references to these standard techniques, focusing and clarifying the changes and specific contribution to this research. Technical details, such as network architecture, should not be part of the introduction and moved to appropriate section in the paper.

The main problem with the paper is that the authors seem to lack some of the existing recent work in the field.  The authors can take a look at the following paper and references therein to consider more recent results: "Comparison and Analysis of Deep Audio Embeddings for Music Emotion Recognition. Eunjeong Koh & Shlomo Dubnov, AAAI 2021: AAAI Workshop on Affective Content Analysis 2021"

Also the comparison in table 3 lacks reference. Are the alternative networks implemented and tested by the authors or quoted from Saari's paper?  This is a relatively old paper, so definitely more updated references are required.  Moreover, the results are only considered in terms of Accuracy, which might be less informative then Precision and Recall or other metrics that are common in information retrieval.

Please note that the name of Saari has typos in the paper (Sarri).

The relation of the classification to Russel's circumflex (Fig 1) is also not clear. Are there any specific emotions that are better classified then others?

The authors need to familiarize themselves with more recent works in the field and update their writing and possibly results accordingly.

Author Response

Thank you so much for your interest in my article!
